# Yeast Diversity Associated with the Phylloplane of Corn Plants Cultivated in Thailand

**DOI:** 10.3390/microorganisms8010080

**Published:** 2020-01-07

**Authors:** Parichat Into, Ana Pontes, José Paulo Sampaio, Savitree Limtong

**Affiliations:** 1Department of Microbiology, Faculty of Science, Kasetsart University, Bangkok 10900, Thailand; parichat_into@hotmail.com; 2UCIBIO-REQUIMTE, Departamento de Ciências da Vida, Faculdade de Ciências e Tecnologia, Universidade Nova de Lisboa, 2829-51 Caparica, Portugal; ap.pontes@fct.unl.pt; 3Academy of Science, The Royal Society of Thailand, Bangkok 10300, Thailand

**Keywords:** yeast, phylloplane, corn leaf, biodiversity

## Abstract

The ecology and diversity of phylloplane yeasts is less well understood in tropical regions than in temperate ones. Therefore, we investigated the yeast diversity associated with the phylloplane of corn, an economically important crop in Thailand, by a culture-dependent method. Thirty-six leaf samples were collected and 217 yeast strains were isolated by plating leaf-washings. The strains were grouped by PCR-fingerprinting and representative strains were identified by analysis of the D1/D2 domain of the large subunit rRNA gene. In total, 212 strains were identified within 10 species in the Ascomycota and 32 species in the Basidiomycota. Five strains represented potential new species in the Basidiomycota, one strain was recently described as *Papiliotrema plantarum,* and four strains belonged to the genera *Vishniacozyma* and *Rhodotorula*. A higher number of strains in the Basidiomycota (81.6%) was obtained. *Hannaella sinensis* was the species with the highest occurrence. Principal coordinates analysis ordinations of yeast communities revealed that there were no differences in the similarity of the sampling sites. The estimation of the expected species richness showed that the observed species richness was lower than expected. This work indicated that a majority of yeast associated with the phylloplane of corn plant belongs to the phylum Basidiomycota.

## 1. Introduction

The phylloplane corresponds to the surface or aboveground parts of plants [1]. Since leaves are the main constituent of the aerial portion of plants, they are viewed as an equivalent of the term phylloplane [2,3]. The phylloplane has been recognized as an important habitat for microorganisms, including bacteria, yeasts and filamentous fungi. The growth of phylloplane microorganisms is dependent on organic substances from plant exudates and inorganic substances present on the leaf surface [4,5]. Moreover, the microbial communities of the phylloplane are dependent of several factors like environmental factors, leaf age, particular growing conditions of the plant, and on the type and density of competing microorganisms [3,6,7]. The phylloplane of plants is colonized by several yeasts of the phyla Ascomycota and Basidiomycota [8,9,10,11,12,13,14,15,16,17,18,19,20,21,22].

In Thailand, rice, sugarcane and corn are crops of major economic relevance used for animal feed (Office of Agriculture Economics, Ministry of Agriculture and Cooperatives of Thailand). The corn cultivation area in 2016 and 2017 was approximately 1.0 and 1.1 million hectares, respectively, and corresponded to the production of 4.1, and 4.8 million tons of corn, respectively. A better knowledge of the corn microbiome, which includes its phylloplane, is of relevance and could have practical implications, such as allowing the design of environmentally friendly biocontrol strategies against corn most relevant pathogens.

The phylloplane yeast community has been studied by both culture-dependent and culture-independent approaches. Recent studies addressed the topic of yeast diversity on the phylloplane of economically relevant crops and both culture-dependent and culture-independent approaches have been employed [13,14,15,16,17,18,19,20,21,22]. As observed for other microbes, the species composition of the yeast communities is not identical when the two approaches are compared or when culture-dependent approaches employ different yeast isolation techniques [16,17,18,19,20,21,23].

There are some reports of phylloplane yeasts in tropical regions and for Thailand the number of such studies has increased substantially in recent years. The phylloplane yeast diversity of the three crops mentioned above (rice, sugarcane and corn) was assessed by culture-dependent and culture-independent approaches [19,20,21,22,23]. However, the assessment of corn phylloplane yeasts was performed only by a culture-independent approach [22]. For rice phylloplane, a culture-dependent approach using enrichment isolation technique was performed [19] and a higher number of species of the Ascomycota was detected, by comparison with representatives of the Basidiomycota. This was in contrast with an assessment by a culture-independent approach that yielded the opposite results [20].

The particular environmental factors operating in different geographic and climatic regions can affect the microbial communities of the phylloplane [24]. Independent studies have suggested that the availability of nutrients, which is dependent on leaf age and growing conditions, can be of relevance for the composition of the yeast communities [3,6,7,25]. Glushakova and Chernov [26] also reported the seasonal changes in the number and species composition of yeast communities in the phylloplane.

The objective of this study was to assess the diversity of yeasts in an economically important crop, the phylloplane of corn plants in Thailand, by using a culture-dependent approach consisting of plating leaf washings for yeast isolation and subsequent molecular identification. Yeast strains were grouped by PCR-fingerprinting with minisatellite-specific oligonucleotides derived from the core sequence of bacteriophage M13, and representative strains of each profile were identified by analysis of the sequence of the D1/D2 domain of the large subunit (LSU) rRNA gene and the complete internal transcribe spacer (ITS) region.

## 2. Materials and Methods

### 2.1. Samples and Yeast Isolation

A total of 36 samples of green and healthy leaves of corn (*Zea mays* Linn.) for animal feed were collected in cultivation fields from 15 districts (sampling sites) in four provinces in Thailand (Figure 1) between July 2016 and October 2016 (Table 1). Each sample was a composite sample consisting of at least five leaves collected from different corn plants in a small area. Leaves were kept in plastic bags and stored at 8 °C and yeast isolation was carried out within 3 days after collection.

Yeasts were isolated by plating of leaf washings as described by Surussawadee et al. [27]. Each composite sample (3 g from all the collected leaves in each sample) was suspended in 50 mL of sterile 0.85% saline solution in a 250 mL Erlenmeyer flask and shaken on a rotary shaker at 25 °C for 1 h to detach yeast cells from leaf surface. An aliquot of 0.1 mL of the washing solution was then spread on yeast extract–malt extract (YM) agar (0.3% yeast extract, 0.3% malt extract, 0.5% peptone and 2.0% agar) supplemented with 0.025% sodium propionate and 0.02% chloramphenicol to inhibit filamentous fungi and bacteria, respectively. The plates were incubated at 20 and 25 °C until yeast colonies appeared. For each sample, three replicates were investigated and yeast colonies of different morphologies were picked and purified by cross-streaking on YM agar. For each sample, similar yeast colonies grown at 20 °C and 25 °C were treated as one strain. Purified yeast strains were suspended in YM broth supplemented with 10% *v*/*v* glycerol and maintained at −80 °C.

### 2.2. Yeast Identification

DNA extraction and purification followed the protocols of Yurkov et al. [28,29]. In brief, DNA was isolated from 3–4 days old cultures. The cells were extracted by the combined physical and chemical methods. The cells were mixed with glass beads and stored at −20 °C for at least one hour and Tris-ethylene-diamine tetracetic acid (TE)-phenol and chloroform was added. The cells were disrupted by vortexing for 20–25 min and centrifuged at 4 °C. The DNA was purified by ethanol precipitation and was then dissolved in 50 µL of TE buffer containing 10 µg/mL RNAse (Sigma-Aldrich, St. Louis, MO, USA).

PCR-fingerprinting with minisatellite-specific oligonucleotides (STAB Vida Inc., Lisbon, Portugal) derived from the core sequence of bacteriophage M13 was used to group yeast strains [30]. Strains showing identical electrophoretic profiles were considered as conspecific and one to two representatives of each PCR fingerprinting group were subjected to sequencing of the D1/D2 domain and ITS region. DNA fragments were amplified by PCR using the primers ITS1f and NL4 [31,32]. The PCR condition were as described by Yurkov et al. [29]. PCR products were purified with GenElute PCR Clean-Up Kit (Sigma-Aldrich, USA) and the purified products were submitted to STAB Vida Inc. (Portugal) for Sanger dideoxy sequencing with ABI 3730 xl sequencer (Applied Biosystems, Foster City, CA, USA).

For yeast identification, the nucleotide sequences obtained in this study were compared with sequences deposited in the NCBI (www.ncbi.nih.gov) and CBS (www.westerdijkinstitute.nl) databases. For identification of ascomycetous yeasts, strains with 0–3 nucleotide differences in the D1/D2 domain were treated as conspecific and strains showing greater than 1% nucleotide substitutions were considered to be different species [33]. When necessary, the “potential new species” designation was used. For the identification of basidiomycetous yeasts, strains differing by two or more nucleotide substitutions were considered to represent different species. When necessary, complete ITS sequences were also analyzed following the previous study [34] in order to assist the D1/D2-based identifications.

### 2.3. Phylogenetic Analysis

Phylogenetic analyses based on the sequences of the D1/D2 domain of the LSU rRNA gene were used to confirm the yeast identification results. Sequences of the type strains of relevant species were obtained from the NCBI database. The sequences were aligned with MUSCLE software version 3.6 [35], provided in MEGA version 7.0 [36]. A phylogenetic tree was constructed from the estimated evolutionary distance data using the general time reversible (GTR) model and the Maximum likelihood method and was performed in MEGA version 7.0. Bootstrap values were determined from 1000 random re-samplings [37].

### 2.4. Biodiversity Analyses

The measure of similarity by the classical Jaccard similarity coefficient (based on species absence or presence) was used in order to assess the similarities of the yeast communities in the samples collected in the 15 sampling sites. The calculation of the Jaccard similarity coefficient (Jaccard index) was performed using the PAST software version 3.25 [38]. The Principal Coordinates Analysis (PCoA) for the ordination of yeast communities in the 15 sampling sites was based on the Jaccard similarity index and employed the PAST software version 3.25 [38]. Species accumulation curves were calculated with EstimateS 9.1.0 using 100 randomizations. Three estimator of species richness were used i.e., Chao1 richness estimator, incidence-based coverage (ICE) estimators and bootstrap richness estimator with sample-based abundance data [39].

## 3. Results

### 3.1. Sample Collection and Yeast Isolation

From 36 leaf samples of corn for animal feed collected in cultivation fields in four provinces, Suphan Buri (*n* = 9), Chai Nat (*n* = 9), Nakhon Sawan (*n* = 9) and Phetchabun (*n* = 9) in Thailand, 217 yeast strains were obtained. As shown in Table 1 similar numbers of yeast strains were obtained in the different provinces, *viz.* 63, 56, 52 and 46 strains obtained from Suphan Buri, Chai Nat, Nakhon Sawan and Phetchabun, respectively.

### 3.2. Yeast Identification

PCR-fingerprinting with primer M13 of 217 strains yielded 101 profiles. From these profiles, 152 strains were selected for identification by sequencing the D1/D2 domain of the LSU rRNA gene and in some cases, the complete ITS region sequence was also used (Appendix A). The phylogenetic placement of the phylloplane strains is shown in Figure 2 (Ascomycota) and Figure 3 (Basidiomycota). The phylloplane yeasts were found to represent 45 species distributed in four main fungal lineages: Ascomycota, Saccharomycotina (10 species, 40 strains), Basidiomycota, Agaricomycotina (16 species, 80 strains), Basidiomycota, Pucciniomycotina (12 species, 48 strains) and Basidiomycota, Ustilaginiomycotina (7 species, 49 strains) (Table 2 and Appendix A, Figure 2 and Figure 3). Among the 45 species found, two species of the *Rhodotorula* (two strains) and *Vishniacozyma* (two strains) in the Phylum Basidiomycota are not yet formally described and therefore represent new species (Appendix A). The known species of Basidiomycota were assigned to 15 genera: *Anthracocystis* (one species), *Cystobasidium* (one species), *Dioszegia* (one species), *Dirkmeia* (one species), *Hannaella* (six species), *Kwoniella* (two species), *Moesziomyces* (one species), *Papiliotrema* (four species), *Rhodosporidiobolus* (three species), *Rhodotorula* (one species), *Saitozyma* (one species), *Sporobolomyces* (four species), *Symmetrospora* (one species) and *Ustilago* (three species) (Table 2 and Appendix A). *Papiliotrema plantarum* was recently described by us [40]. The ten Ascomycota species belonged to six genera: *Candida* (four species)*, Kodamaea* (one species)*, Meyerozyma* (one species)*, Metschnikowia* (one species)*, Pichia* (two species) and *Wickerhamomyces* (one species) (Table 2 and Appendix A). According to the BLASTn search, the D1/D2 and ITS sequences of these four strains were similar to several sequences of unidentified species that are deposited in the GenBank database. The complete characterization of these sequences and strains will be necessary before the formal description of new species is proposed. The number of yeasts of the phylum Basidiomycota (81.6%) was much higher than those assigned to the Ascomycota (18.4%).

### 3.3. Yeast Diversity

Considering the strains isolated in this study, the number of yeasts of the phylum Basidiomycota (81.6%) was much higher than those assigned to the Ascomycota (18.4%). *Hannaella sinensis* was the species with the highest occurrence species, it was found in 24 samples out of 36 samples (66.7%). The comparison of the similarity of the yeast community from each sampling sites was carried out in pairwise comparisons. It showed that the yeast species composition was different among all sampling sites. The similarity coefficient values were in the range of 0–0.57 (Appendix A). Globally we obtained an average value of 0.25 for the similarity between yeast communities, which could mean that the sampling sites shared, on average 25% of the species. A comparatively higher similarity (0.57) was observed between sampling site 14 (Mueang Phetchabun, Phetchabun, Thailand) and sampling site 15 (Chon Daen, Phetchabun, Thailand). On the opposite side, the comparison of similarity of the sampling site 4 (Si Prachan, Supahn Buri, Thailand) and sampling site 7 (Manorom, Chai Nat, Thailand) had a similarity index = 0, and the same happened for the comparison between sampling site 7 (Manorom, Chai Nat, Thailand) and sampling site 11 (Tak Fa, Nakhon Sawan, Thailand). However, the PCoA plot based on Jaccard similarity indices suggested that the yeast communities of all sampling sites were in the same group and that there were no marked differences on the similarity on the sampling sites (Figure 4). The estimation of the expected species richness taking in consideration the sampling effort by the Bootstrap, Chao 1 and ICE estimators revealed that the observed species richness was lower than the expected species richness (Figure 5).

## 4. Discussion

In this work, we employed a culture-dependent approach consisting on plating leaf washings to analyze the yeast community of corn phylloplane. We measured a much larger proportion of basidiomycetous yeasts (81.6%) than the fraction of ascomycetous yeasts. This result is in accordance with our previous study of yeasts from the corn phylloplane using a culture-independent approach that showed that the majority (98.5%) of yeast sequences belonged to the Basidiomycota [22]. Moreover, this result is in agreement with other studies that employed the culture-dependent approach, like that of de Azeredo et al. [8], who found that most of the detected yeast species belonged to the Basidiomycota (92.2%). However, we found that species composition of the yeast community associated with corn phylloplane inferred with culture-dependent and culture-independent approaches is different. The same conclusion can be obtained in the comparison of results concerning analyses of the rice yeast community in Thailand [21,23].

In our study, basidiomycetous yeasts were found to belong to the three main lineages of the Basidiomycota: the Pucciniomycotina, Ustilaginomycotina and Agaricomycotina. The species with the highest occurrence species was *Hannaella* (*Bullera*) *sinensis*. This species was first described based on a strain obtained from a wheat leaf (*Triticum* sp.) in China [41] and later found to occur widely in the tissue and the leaf surface of various plants such as rice, corn, sugarcane, elephant grass, and cactus [20,42,43,44,45,46,47]. *Dirkmeia churashimaensis*, *P. flavescens*, *P. rajasthanensis* and *R. paludigena*, which were frequently found in this study, were also commonly found in the phylloplane of trees [17,48,49]. The species with the highest occurrence species found in this study (*H. sinensis*) is distinct from the most abundant species found in a culture-independent survey, that reported *Pseudozyma hubeinsis* pro tem. and *Moesziomyces antarcticus* as the species with the highest occurrence yeasts [22]. Moreover, 31 species detected in this study were not found when a culture-independent method was used (e.g., *M. caribbica*, *P. flavescens*, *P. rajasthanensis* and *R. paludigena*). On the other hand, 12 species found by a culture-independent method were not detected in this study (e.g., *Moesziomyces aphidis*, *Meyerozyma rugulosus*, *Meyerozyma guilliermondii*, *Pseudozyma hubeiensis* and *Spencerozyma crocea*) [22]. Although these two investigations were carried out in Thailand, they were performed in different seasons and in different geographic locations. In the present study, we collected the samples during the rainy season (June to October), while Nasanit et al. [22] collected samples between January and May, during the dry season. Such different sampling strategies might partly explain the different results. Also, the two studies were not performed in the same provinces, except for Suphan Buri and Nakorn Swan. de Azeredo et al. [8] assessed the diversity of phylloplane yeasts in sugarcane leaf in Brazil by plating of leaf-washings and found as prevalent species *Debaryomyces hansenii, Naganishia albidus* (formerly *Cryptococcus albidus*), *Papiliotrema laurentii* (formerly *Cryptococcus laurentii*) and *Rhodotorula mucilaginosa*, which differs considerably from the prevalent community found in corn in our study.

Another aspect previously documented [22,24,26] and supported by our studies, concerns the marked differences obtained in different seasons and geographic locations. It is relevant to note that in our study we have collected corn leaf samples in different growth stages and growing conditions of the plants, which may have affected the results. Also, the sampling site climatic condition, which relates with the sampling period of the study, leaf age and technique of investigation involved in the analysis of the yeast communities are also critical factors that need to be taken into consideration. Therefore, although our study illuminates phylloplane yeast diversity in corn leaves in a tropical region, further studies should aim at clarifying if species richness is affected by sampling locations, growing conditions and leaf age.

We obtained discrepant results between the comparison of species compositions among the sampling sites using the Jaccard similarity coefficient and the analysis using a PCoA plot. This result suggests that there is no significant spatial heterogeneity in what concerns the composition of the phylloplane yeast community. Moreover, since in this study we collected the samples only in the central part of Thailand, additional sampling in other locations should clarify the relationship between the yeast community composition and geography. The observed species richness was lower than the expected species richness. Overall, species richness estimators perform differently depending on the yeast community structure. An increase in the number of rare species results in higher species richness values predicted by the estimators (Bootstrap, Chao 1 and ICE), that distinguish between rare and frequent species. Our results suggest that rare yeast species were present in the community and that they remained unobserved. This result, and the failure to fully document the existing yeast diversity was also obtained in other similar studies [20,21,22,50]. It is possible that a more exhaustive sampling could have led to a more precise estimate of the actual yeast diversity indices.

Besides documenting the yeast diversity of corn phylloplane our study provides additional data to support the view that the technique of investigation involved in the analysis of the yeast communities, the choice of sampling sites, the climatic condition, which might relate to the sampling period of the study, and the plant species, all are critical factors that need to be taken into consideration when investigating the yeast diversity associated with the phylloplane.

In recent years, yeasts isolated from various natural habitats have been found to havepotential as effective producers of various biotechnological products worldwide. Various wild yeast species have been reported to have a vital role in modern industrial biotechnology, for example, for the production of citric acid [51], single cell proteins [52,53], ethanol [54,55], microbial oils [56,57,58], indole-3- acetic acid (IAA) [59], itaconic acid [60], glycolipid biosurfactants [49], arabitol [61], xylitol [62], erythritol and mannitol [63]. In this study, we isolated 217 yeast strains of 10 species in the Ascomycota and 35 species in the Basidiomycota. Some of these species have been reported to have industrial biotechnological potential. *Candida intermedia*, *Candida tropicalis* and *Pichia kudriavzevii* have been reported to produce ethanol and single-cell protein [52,53,54,55] whereas *Wickerhamomyces anomalus* has been found to have the ability to produce ethanol and xylitol [54,62]. *Meyerozyma caribbica* has exhibited potential for xylitol production [64]. Various species in the Basidiomycota found in this study including *P. laurentii, R. paludigena, Rhodotorula toruloides, Rhodosporidiobolus fluvialis*, *Rhodosporidiobolus ruineniae* and *Sporobolomyces carnicolor* have been reported for their potential in microbial lipid production [56,57,58]. *Papiliotrema flavescens* has been reported to be able to produce α- galactosidase [65]. The strains of *R. paludigena* and *H. sinensis* also showed the ability to produce large qualities of indole-3-acetic acid (IAA) [59]. *D. churashimaensis* and *M. antarcticus*, which previously belonged to the genus *Pseudozyma*, have been applied for itaconic acid production [60]. Moreover, these two species and *Ustilago (Pseudozyma) siamensis* have been found to be capable of producing a mixture of mannosylerythritol lipids, which are glycolipid biosurfactants [49]. These findings suggest that some of the yeast strains isolated in this study could have potential for production of various industrial biotechnological products.

## 5. Conclusions

The result of this study revealed that a majority of yeasts on the phylloplane of the corn plant were in the phylum Basidiomycota when assessed by a culture-dependent method using the plating of leaf washings for yeast isolation, which was in accordance with assessment done by a culture-independent method. The finding of that basidiomycetous yeasts are dominant on the phylloplane of the corn plant is similar to what is found in other plant species. The yeast species with the highest occurrence was *H. sinensis*. Some species of corn phylloplane yeasts found in this study differed from those found in a study using a culture-independent method due to differences in the sampling period and to intrinsic biases of the two methods. To obtain a better characterization of the yeast community inhabiting the phylloplane, we suggest using a culture-dependent method based on direct isolation and employing different culture media in combination with a culture-independent approach. In addition, some of the yeast strains obtained in this study could have potential for application in industrial biotechnology.

## Figures and Tables

**Figure 1 microorganisms-08-00080-f001:**
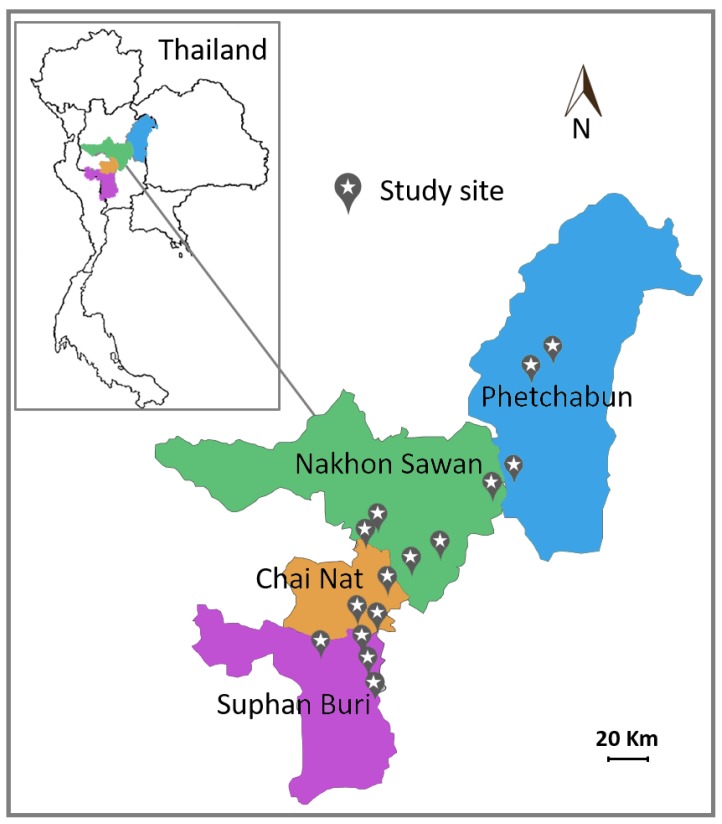
Map of sampling sites in four provinces in Thailand.

**Figure 2 microorganisms-08-00080-f002:**
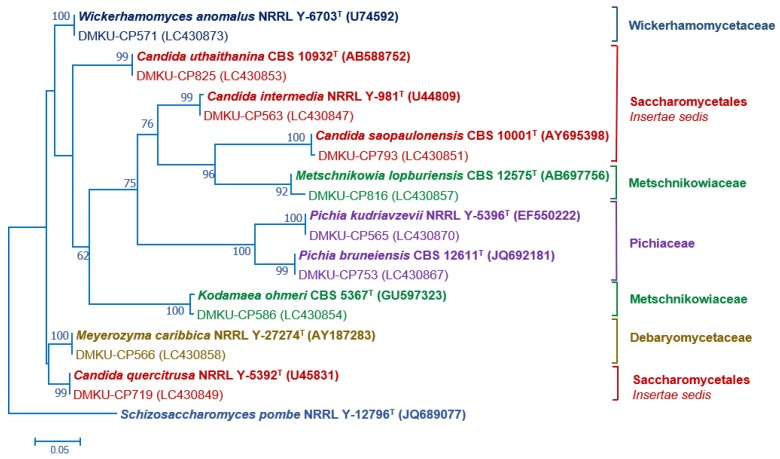
Phylogenetic placement of known species of corn phylloplane ascomycetous yeasts (Phylum Ascomycota, subphylum Saccharomycotina) based on the sequence of the D1/D2 region of LSU rRNA gene. Reference sequences retrieved from the GenBank database are included. The tree was constructed with the maximum-likelihood method and the GTR evolutionary model. Numbers on the branches represent the bootstrap values (>50%) from 1000 random replicates. The scale bar corresponds to a genetic distance of 0.05 substitutions per position.

**Figure 3 microorganisms-08-00080-f003:**
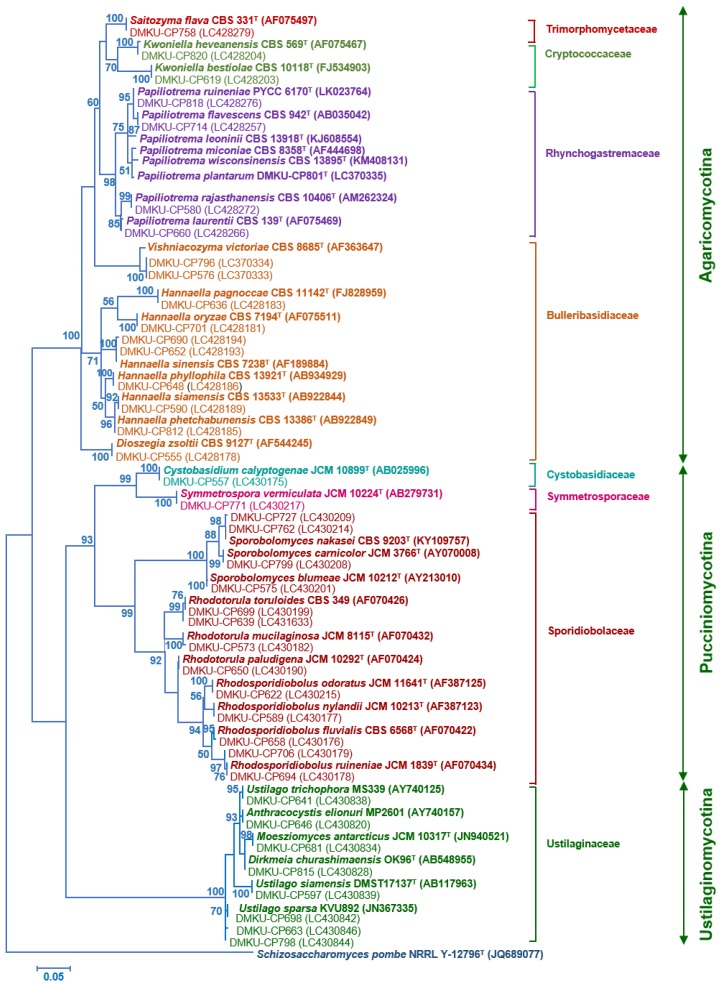
Phylogenetic placement of known species of corn phylloplane basidiomycetous yeasts (Phylum Basidiomycota: subphylum Pucciniomycotina, Ustilaginomycotina and Agaricomycotina) based on the sequence of the D1/D2 region of LSU rRNA gene. Reference sequences retrieved from the GenBank database are included. The tree was constructed with the maximum-likelihood method and the GTR evolutionary model. Numbers on branches represent the bootstrap values (>50%) from 1000 random replicates. The scale bar corresponds to a genetic distance of 0.05 substitutions per position.

**Figure 4 microorganisms-08-00080-f004:**
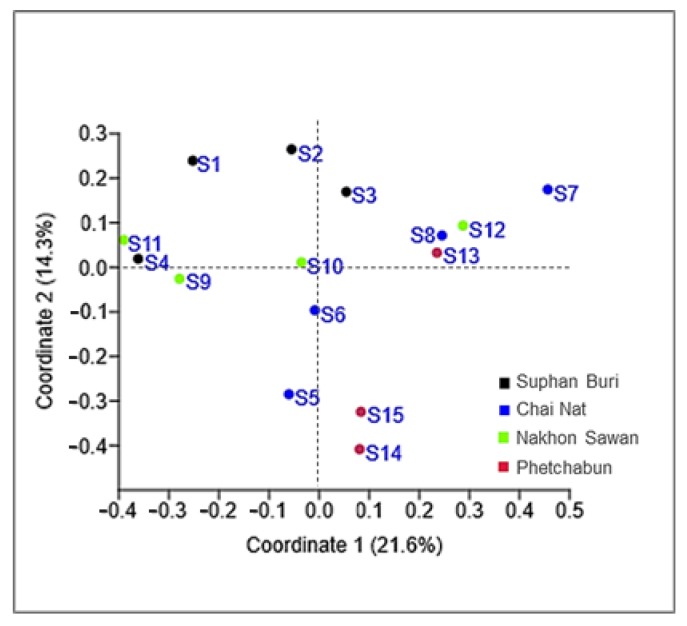
Principal Coordinate Analysis (PCoA) plots of corn phylloplane yeast communities in 15 sampling sites in Thailand, using the Jaccard similarity coefficient.

**Figure 5 microorganisms-08-00080-f005:**
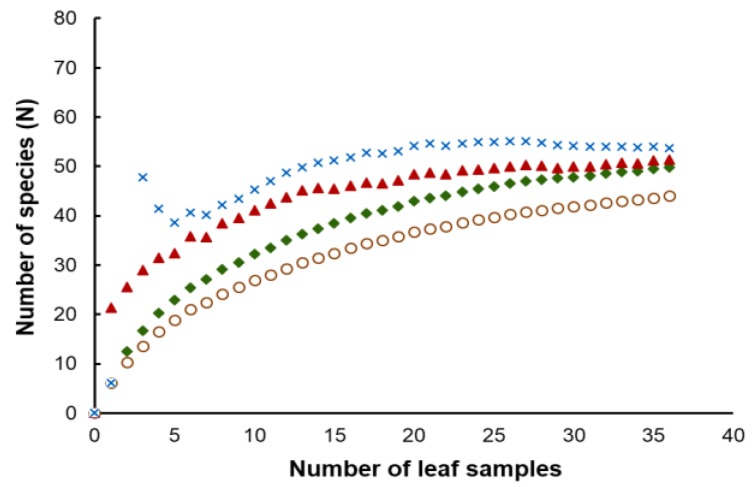
Observed (unfilled circle) curves and estimated phylotype richness of corn phylloplane yeast versus sample size using bootstrap (filled diamond), Chao 1 (filled triangle) and ICE (cross) estimators using sample-based abundance data.

**Table 1 microorganisms-08-00080-t001:** Corn leaf samples and correspondent number of strains isolated during this study.

	Sampling					No. of Sample	No. of Strain
Site	Province District	Location	Date (dd mm yy)	Atmospheric Temperature ^a^ (°C)	Rainfall ^b^ (mm)		
	Suphan Buri						
S1	Doembangnangbuat	14°50′27.1″ N 100°07′51.5″ E	2016-07-18	29.2–29.8	100.2–157.0	2	19
S2	Sam Chuk	14°47′18.4″ N 100°09′14.1″ E	2016-07-30			3	22
S3	Nong Ya Sai	14°49′22.6″ N 99°55′46.0″ E	2016-08-06			2	9
S4	Si Prachan	14°38′36.8″ N 100°08′53.1″ E	2016-08-06			2	13
					Total	9	63
	Chai Nat						
S5	Hankha	14°59′55.3″ N 100°05′45.6″ E	2016-08-06	28.1–28.6	208.8–358.6	4	22
S6	Sankhaburi	14°57′37.8″ N 100°10′57.6″ E	2016-08-06			2	17
S7	Manorom	15°23′11.6″ N 100°08′29.6″ E	2016-08-13			1	4
S8	Sapphaya	15°08′15.5″ N 100°14′44.7″ E	2016-09-17			2	13
					Total	9	56
	Nakhon Sawan						
S9	Phayuha Khiri	15°28′06.3″ N 100°12′17.7″ E	2016-09-16	28.1–28.6	374.7	4	19
S10	Phaisali	15°38′20.6″ N 100°46′49.6″ E	2016-09-16			2	16
S11	Tak Fa	15°21′21.2″ N 100°30′10.0″ E	2016-09-17			2	12
S12	Takhli	15°16′19.5″ N 100°21′48.1″ E	2016-09-17			1	5
					Total	9	52
	Phetchabun						
S13	Bueng Sam Phan	15°43′51.0″ N 100°52′34.9″ E	2016-09-16	27.2–27.8	290.5	2	10
S14	Mueang Phetchabun	16°18′08.2″ N 101°03′58.2″ E	2016-09-17			3	15
S15	Chon Daen	16°12′32.6″ N 100°57′48.9″ E	2016-09-17			4	21
					Total	9	46

^a^ Range of atmospheric temperature obtained in particular month of sampling area. ^b^ Range of rainfall obtained in particular month of sampling area.

**Table 2 microorganisms-08-00080-t002:** Yeast species and respective frequencies detected on corn phylloplane.

Taxa	Number of Strain				Total	FO (%) ^a^
	Suphan Buri	Chai Nat	Nakhon Sawan	Phetchabun		
**Phylum Ascomycota** **Subphylum Saccharomycotina**						
*Candida intermedia*	1	1	3	-	5	13.9
*Candida quercitrusa*	-	-	-	1	1	2.8
*Candida saopaulonensis*	-	2	-	2	4	11.1
*Candida uthaithanina*	-	1	-	-	1	2.8
*Kodamaea ohmeri*	1	-	-	-	1	2.8
*Metschnikowia lopburiensis*	-	1	1	1	3	8.3
*Meyerozyma caribbica*	2	3	5	4	14	38.9
*Pichia bruneiensis*	-	2	-	4	6	16.7
*Pichia kudriavzevii*	1	-	1	-	2	5.6
*Wickerhamomyces anomalus*	2	1	-	-	3	8.3
**Phylum Basidiomycota** **Subphylum Agaricomycotina**						
*Dioszegia zsoltii*	1	1	1	-	3	8.3
*Hannaella oryzae*	-	-	2	-	2	5.6
*Hannaella pagnoccae*	-	2	-	1	3	8.3
*Hannaella phetchabunensis*	-	1	-	-	1	2.8
*Hannaella phyllophila*	-	1	1	1	3	8.3
*Hannaella siamensis*	1	2	-	2	5	13.9
*Hannaella sinensis*	5	8	6	5	24	66.7
Potential new species closest to *Vishniacozyma heimaeyensis*	1	1	-	-	2	5.6
*Kwoniella bestiolae*	1	-	-	-	1	2.8
*Kwoniella heveanensis*	-	1	-	-	1	2.8
*Papiliotrema flavescens*	1	5	2	6	14	38.9
*Papiliotrema laurentii*	3	1	1	-	5	13.9
*Papiliotrema rajasthanensis*	5	2	3	1	11	30.6
*Papiliotrema ruineniae*	-	1	1	-	2	5.6
*Plapiliotrema plantarum*	-	1	-	-	1	2.8
*Saitozyma flava*	1	-	-	1	2	5.6
**Subphylum Pucciniomycotina**						
*Cystobasidium calyptogenae*	1	-	-	-	1	2.8
*Symmetrospora vermiculata*	1	1	1	-	3	8.3
*Rhodotorula mucilaginosa*	2	-	-	-	2	5.6
*Rhodotorula paludigena*	5	4	5	4	18	50.0
*Rhodotorula toruloides*	-	-	1	-	1	2.8
Potential new species closest to *Rhodotorula toruloides*	1	1	-	-	2	5.6
*Rhodosporidiobolus nylandii*	1	-	-	-	1	2.8
*Rhodosporidiobolus ruineniae*	-	-	2	1	3	8.3
*Rhodosporidiobolus fluvialis*	-	1	-	-	1	2.8
*Rhodosporidiobolus odoratus*	1	-	-	-	1	2.8
*Sporobolomyces blumeae*	5	-	2	-	7	19.4
*Sporobolomyces nakasei*	-	-	-	6	6	16.7
*Sporobolomyces carnicolor*	-	1	-	1	2	5.6
**Subphylum Ustilaginomycotina**						
*Anthracocystis elionuri*	1	3	2	-	6	16.7
*Dirkmeia churashimaensis*	8	4	4	1	17	47.2
*Moesziomyces antarcticus*	3	-	5	2	10	27.8
*Ustilago trichophora*	1	1	-	-	2	5.6
*Ustilago siamensis*	1	-	-	-	1	2.8
*Ustilago sparsa*	6	2	3	2	13	36.1

^a^ FO; Frequency of occurrence (%) was calculated as the number of samples, where a particular species was observed, as a proportion of the total number of samples.

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
