# Peer review of "Yeast Diversity Associated with the Phylloplane of Corn Plants Cultivated in Thailand"

_microorganisms, 2020, doi:10.3390/microorganisms8010080_

Round 1

Reviewer 1 Report

This paper deals with the yeast diversity of corn leaves in Thailand, using a culture-dependent approach. It brings relevant data regarding yeast diversity.

English in the abstract is poor when compared to the rest of the paper; authors should review. See for instance:

Line 13

“There were smaller number of report”

Line 14

“Therefore, we investigate”

Line 20

“which”

Line 21

“were belonged”

Line 22

“was detected”

Line 23

“The most occurrence”

Other comments

Lines 24-27

Is this the main finding and relevance of the work? Authors could end the abstract with the significance of the work.

Line 51

“identical when the two”; when is missing

Line 77

Why does the number of samples from each site is different? It is the same for each province (n=9), but the number in each site varies from 2-4. 

Line 99

“and of with”, please correct

Line 99

“TE-phenol” – TE should be in full also, it is the first time the abbreviation appears; please include what it stands for in full. 

Line 99

“by Vortex mixture”-> by vortexing

Line 101

Perhaps the volume, concentration and source of the RNAase should be mentioned.

Line 103

Source (e.g. vendor/supplier) of the minisatellite-specific oligonucleotides?

Line 104

“and one to two representatives of each group PCR fingerprinting groups”; Confusing, please review.

Line 106

“The PCR condition” -> The PCR conditions were as described…

Line 108

What was the methodology/equipment used for sequencing?

Line 139

“estimators”-> estimator

Line 152

“phylogenic” -> phylogenetic

Line 174

“based the sequence” -> based on the sequence; same for the legend of Figure 3

Line 176

maximum-Likelihood method -> maximum-likelihood method; same for the legend of Figure 3

Table 2

Potential new species closet to -> closest (correct in both instances, this phrase appears twice in the table)

Line 188

“calculated as a number” -> “calculated as the number”

Lines 193; 232; 237; 239

“…was the most occurrence”-> was the species with the highest occurrence

Since Figure 4 represents the data of Table 3, is Table 3 really necessary? Perhaps it could be moved to Supplementary Material

Line 221-229

There appears to be contradiction. Authors refer the results of culture dependent and culture-independent approaches are similar, but then say it is of relevance that they are different.

Line 223

98.5 % -> 98.5% (omit the space)

Line 281

Authors could highlight the importance of the results in the conclusion.

Author Response

Response to the reviewers #1’ comments

Manuscript ID: microorganisms-677725:

Yeast diversity associated with the phylloplane of corn plants cultivated in Thailand

Comments and Suggestions for Authors

This paper deals with the yeast diversity of corn leaves in Thailand, using a culture-dependent approach. It brings relevant data regarding yeast diversity.

Point 1: English in the abstract is poor when compared to the rest of the paper; authors should review. See for instance:

Response1: For the revised version, we carefully reviewed and also asked native English speaker to check English of the abstract.

Point 2; Line 13: “There were smaller number of report”

Response 2: The sentence contained these words was change to “The ecology and diversity of phylloplane yeasts is less well understood in tropical regions than in temperate ones.”

Point 3; Line 14: “Therefore, we investigate”

Response 3: changed to “Therefore, we investigated”.

Point 4; Line 20: “which”

Response 4: deleted “which” .

Point 5; Line 21:  “were belonged”

Response 5: changed to “belonged”.

Point 6; Line 22: “was detected”

Response 6: changed to “was obtained”.

Point 7; Line 23: “The most occurrence”

Response7: changed “The most occurrence species was Hannaella sinensis” to “Hannaella sinensis was the species with the highest occurrence”.

Other comments

Point 8; Lines 24-27: Is this the main finding and relevance of the work? Authors could end the abstract with the significance of the work.

Response 8: Revised to add the main finding of the work from “The estimation of the expected species richness showed that the observed species richness was lower than expected, which suggests that the real diversity of corn phylloplane yeast is higher than that documented here.” to “The estimation of the expected species richness showed that the observed species richness was lower than expected. This work indicated that the majority of yeasts associated with the phylloplane of corn plants bolongs to the phylum Basidiomycota.”.

Point 9; Line 51: “identical when the two”; when is missing

Response 9: added “when”.

Point 10; Line 77: Why does the number of samples from each site is different? It is the same for each province (n=9), but the number in each site varies from 2-4.

Response 10: We collected 9 samples from each province by randomly collected the sample from each district of the province (sampling site). The samples were collected from corn fields along the road and the sampling site should far from the other at least 2 km. The districts have different area and some are small, so not many corn field that we can collected the sample. Therefore, the samples from each site is different.

Point 11; Line 99: “and of with”, please correct

Response11: corrected to “with”.

Point 12; Line 99: “TE-phenol” – TE should be in full also, it is the first time the abbreviation appears; please include what it stands for in full.

Response12: changed to “Tris-ethylene-diamine tetracetic acid (TE)-phenol”.

Point 13; Line 99: “by Vortex mixture”-> by vortexing

Response13: changed to “by vortexing”.

Point 14; Line 101: Perhaps the volume, concentration and source of the RNAase should be mentioned.

Response14: We added the volume, concentration and source of the RNAase as ‘The DNA was purified by ethanol precipitation and was then dissolved in 50 µl of TE buffer containing containing10 µg/ml RNAse (Sigma-Aldrich, USA).”.

Point 15; Line 103: Source (e.g. vendor/supplier) of the minisatellite-specific oligonucleotides?

Response 15: We added the supplier of the minisatellite-specific oligonucleotides as STAB Vida Inc., Portugal.

Point 16; Line 104: “and one to two representatives of each group PCR fingerprinting groups”; Confusing, please review.

Response16: change to “and one to two representatives of each PCR fingerprinting group were subjected to sequencing of the D1/D2 domain and ITS region”.

Point 17; Line 106: “The PCR condition” -> The PCR conditions were as described…

Response17: Corrected.

Point 18; Line 108: What was the methodology/equipment used for sequencing?

Response18: We added the methodology/equipment used for sequencing as “Sanger sequencing using ABI 3730 xl sequencers (Applied Biosystems, Foster City, CA, USA).”.

Point 19; Line 139: “estimators”-> estimator

Response19: Corrected.

Point 20; Line 152: “phylogenic” -> phylogenetic

Response 20: Corrected.

Point 21; Line 174: “based the sequence” -> based on the sequence; same for the legend of Figure 3

Response 21: Corrected.

Point 22; Line 176: maximum-Likelihood method -> maximum-likelihood method; same for the legend of Figure 3

Response 22: Corrected.

Point 23; Table 2: Potential new species closet to -> closest (correct in both instances, this phrase appears twice in the table)

Response 23: Corrected.

Point 24; Line 188: “calculated as a number” -> “calculated as the number”

Response 24: Corrected.

Point 25; Lines 193; 232; 237; 239: “…was the most occurrence”-> was the species with the highest occurrence

Response 25: Corrected.

Point 26; Since Figure 4 represents the data of Table 3, is Table 3 really necessary? Perhaps it could be moved to Supplementary Material

Response 26: Moved Table 3 to Supplementary Material.

Point 27; Line 221-229: There appears to be contradiction. Authors refer the results of culture dependent and culture-independent approaches are similar, but then say it is of relevance that they are different.

Response27: We would like to mention that the species composition of the yeast community is different. The sentence “It is of relevance that the composition of the yeast community associated with corn phylloplane inferred with culture-dependent and culture-independent approaches is different.” was changed to “However, we found that species composition of the yeast community associated with corn phylloplane inferred with culture-dependent and culture-independent approaches is different”.

Point 28; Line 223: 98.5 % -> 98.5% (omit the space)

Response28: Corrected.

Point 29; Line 281: Authors could highlight the importance of the results in the conclusion

Response 29: We make some change in the conclusion part.

Reviewer 2 Report

Reference: Microorganisms 677725

Title: Yeast diversity associated with the phylloplane of corn plants cultivated in Thailand

            In a potentially interesting m/s, Into et al investigated the yeast diversity associated with the phylloplane of corn, an economically important crop various countries. In total 212 strains were identified within 10 species in the Ascomycota and 32 species in the Basidiomycota. Five strains represented potential new species in the Basidiomycota, where one strain was recently described as Papiliotrema plantarum and four strains were belonged to the genera Vishniacozyma and Rhodotorula. Other microorganisms isolated and identified belonged to the genera Metschnikowia, Pichia, Sporobolomyces, etc.

            A large quantity of experimental work has been carried out in the present study. The main drawback of the current submission refers to the fact that very limited discussion in relation to the biotechnological potential of several of the new isolates has been presented in the current submission. Unlike the presentations done by the authors, in several types of fermentation processes that are very important for the modern Industrial Biotechnology (i.e. the ones of production of citric acid, single-cell oil amenable to be converted into biodiesel, mannitol, arabitol, ethanol, etc), utilization of wild-type microorganisms isolated from various natural habitats similar with the ones presented in the current submission (i.e. soil, sourdoughs, marine fish, fruits, etc) has resulted in very interesting results that merit to be cited and discussed in parallel with the results obtained here [see: Biomass Bioenerg (2010) 34, 101–107; Marine Biotechnol (2013) 15, 26–36; Eng Life Sci (2017) 17, 333–344; J Appl Microbiol (2019) 127, 1080–1100; Microorganisms (2019) 7, 633]. Therefore, strains of Pichia sp. are capable to produce single-cell protein, strains of Metschnikowia sp. are capable to produce ethanol and/or single-cell oil, strains of the family Debaryomycetaceae are capable to produce arabitol, while various strains of the genera Rhodotorula/Rhodosporidium sp. are capable to produce single-cell oil and pigments; all these microorganisms are capable to grow in various waste- and residue-streams [see and comprehensively discuss: Eng Life Sci (2017) 17, 333–344; J Appl Microbiol (2019) 127, 1080–1100; Microorganisms (2019) 7, 229]. All these points need to be comprehensively discussed by the authors.

            For yeast species, identification of the 5.8S-ITS rDNA region is considered to exhibit the highest resolving power for discriminating closely related fungal species [see: J Appl Microbiol (2019) 127, 1080–1100]. It is not clear whether this method was used in the current submission.

            Moderate revision in the points presented by the referee is requested.

Author Response

Response to the reviewers #2’ comments

Manuscript ID: microorganisms-677725:

Yeast diversity associated with the phylloplane of corn plants cultivated in Thailand

Point 1: In a potentially interesting m/s, Into et al investigated the yeast diversity associated with the phylloplane of corn, an economically important crop various countries. In total 212 strains were identified within 10 species in the Ascomycota and 32 species in the Basidiomycota. Five strains represented potential new species in the Basidiomycota, where one strain was recently described as Papiliotrema plantarum and four strains were belonged to the genera Vishniacozyma and Rhodotorula. Other microorganisms isolated and identified belonged to the genera Metschnikowia, Pichia, Sporobolomyces, etc.

A large quantity of experimental work has been carried out in the present study. The main drawback of the current submission refers to the fact that very limited discussion in relation to the biotechnological new potential of several of the isolates has been presented in the current submission. Unlike the presentations done by the authors, in several types of fermentation processes that are very important for the modern Industrial Biotechnology (i.e. the ones of production of citric acid, single-cell oil amenable to be converted into biodiesel, mannitol, arabitol, ethanol, etc), utilization of wild-type microorganisms isolated from various natural habitats similar with the ones presented in the current submission (i.e. soil, sourdoughs, marine fish, fruits, etc) has resulted in very interesting results that merit to be cited and discussed in parallel with the results obtained here [see: Biomass Bioenerg (2010) 34, 101–107; Marine Biotechnol (2013) 15, 26–36; Eng Life Sci (2017) 17, 333–344; J Appl Microbiol (2019) 127, 1080–1100; Microorganisms (2019) 7, 633]. Therefore, strains of Pichia sp. are capable to produce single-cell protein, strains of Metschnikowia sp. are capable to produce ethanol and/or single-cell oil, strains of the family Debaryomycetaceae are capable to produce arabitol, while various strains of the genera Rhodotorula/Rhodosporidium sp. are capable to produce single-cell oil and pigments; all these microorganisms are capable to grow in various waste- and residue-streams [see and comprehensively discuss: Eng Life Sci (2017) 17, 333–344; J Appl Microbiol (2019) 127, 1080–1100; Microorganisms (2019) 7, 229]. All these points need to be comprehensively discussed by the authors.

Response1: We added discussion in point of your comments as in L 285-305 of the revised MS.

Point 2: For yeast species, identification of the 5.8S-ITS rDNA region is considered to exhibit the highest resolving power for discriminating closely related fungal species [see: J Appl Microbiol (2019) 127, 1080–1100]. It is not clear whether this method was used in the current submission.

Response 2: Both the D1/D2 domain of the LSU rRNA gene and the 5.8S-ITS rRNA gene region sequences have been used for yeast identification. Both genes had potential in discriminating closely related yeast species. Although in the recent years, the sequences of the 5.8S-ITS rDNA region has selected to be a DNA barcode for fungi, however, D1/D2 domain of the LSU rRNA gene is also considered as the DNA barcode of yeasts. The D1/D2 domain of the LSU rRNA gene still more popular for yeast identification among yeast taxonomist, because of there are a larger number of the D1/D2 sequences in database than the ITS sequences. Many of the current papers published in 2019 [such as Microorganisms (2019) 7, 633; Int J Food Microbiol (2019), 305, 108255] still used the sequences of the D1/D2 domain for yeast identification. In the present study, the ITS sequence was used together with the sequence of the D1/D2 domain in the case of new and potential new yeast species.

Round 2

Reviewer 2 Report

 The paper seems correct and can be published in the journal. Please pay attention to some small grammatical errors that still exist.